# New Characteristics in the Fermentation Process of Cocoa (*Theobroma cacao* L.) "Super Árbol" in La Joya de los Sachas, Ecuador

Maritza Sanchez-Capa [1,2,*], Samuel Viteri-Sanchez [3], Armando Burbano-Cachiguango [3],
Mauricio Abril-Donoso [4], Tannia Vargas-Tierras [1], Sandra Suarez-Cedillo [5] and Carlos Mestanza-Ramón [1,2]

1. Research Group YASUNI-SDC, Escuela Superior Politécnica de Chimborazo, Sede Orellana, El Coca 220001, Ecuador; tannia.vargas@espoch.edu.ec (T.V.-T.); cmestanza@us.es (C.M.-R.)
2. Departamento de Ciencias Agroforestales, ETSIA, Universidad de Sevilla, Crta de Utrera Km 1, 41013 Seville, Spain
3. National Institute of Agricultural Research (INIAP), Central Experimental Station of the Amazon (EECA), La Joya de Los Sachas 220101, Ecuador; samuel.viteri@iniap.gob.ec (S.V.-S.); remigio.burbano@iniap.gob.ec (A.B.-C.)
4. Directorate of Statistics and Analysis of Health Information, Ministry of Public Health, Quito 170702, Ecuador; mauricio.abril@msp.gob.ec
5. Escuela Superior Politécnica de Chimborazo, Sede Orellana, El Coca 220001, Ecuador; sandra.suarez@espoch.edu.ec
* Correspondence: maritzac.sanchez@espoch.edu.ec

**Abstract:** In Ecuador, since 2005 in the northern Amazon, trinitario hybrid cacao mother plants characterized by early and abundant fruiting, known as "Super árbol", have been identified. This genetic material was disseminated in the region, but most of the available information corresponds to empirical knowledge. In this sense, the present study aimed to evaluate different fermentation techniques in the "Super árbol" cocoa by analyzing physical and chemical variables in the climatic conditions of the Joya de Los Sachas canton to establish differences between the group of genetic material of the "Super árbol" cocoa with respect to the "Arriba" variety, which is considered a reference in Ecuador. The physical and chemical parameters evaluated were: fermentation rate, weight of 100 beans, pH cotyledon, beans shell, protein, ash, lipid, and total polyphenols. The study was conducted under a completely randomized design with three factors. A Spearman correlation analysis was performed, followed by the establishment of a model for each variable and the use of Tukey's test to establish the difference between means of treatments and a Levene's test to test homogeneity. The "Super árbol" cocoa reported fermentation percentages between 64.33 and 95%, testa percentages between 13.28 and 18.08%, and polyphenol content between 48.46–55.54 GAE/g DW. Thus, this genetic material of the "Super árbol" trinity group has characteristics that reach higher fermentation percentages compared with the "Arriba" variety. In addition, it has a lower polyphenol content (less bitter and astringent), which leads to a better-quality raw chocolate material.

**Keywords:** cocoa "Arriba"; polyphenols; amazon region; pre-dried; chocolate

## 1. Introduction

Cocoa (*Theobroma cacao* L.) is a cash crop with about 8.2 million hectares planted in the world [1]. The main producers are the developing countries located in the tropics, while the main consumers are the developed countries in the temperate zones of the world. Currently, more than 58 countries around the world are engaged in cocoa production [2]. About 89% of cocoa production is supplied by leading countries such as Ivory Coast, Ghana, Cameroon, Nigeria, Ecuador, Brazil, Indonesia, and New Guinea. Globally, total cocoa bean production exceeds 4.7 million tons as recorded in the 2018/2019 cocoa season [3].

In Latin America and the Caribbean, cocoa is a subsector that stimulates territorial development, and activates and mobilizes local economies [4,5]. Among the main characteristics of the beans produced in countries of the region such as Ecuador, Brazil, Peru, Colombia, and Venezuela, it should be noted that they come mainly from cultivars of Trinidadian and Creole origin. These origins stand out for their quality and potent differentiated aroma, characteristics that have allowed them to enter the main specialty markets of the world [1,3,6].

In Ecuador, cocoa is one of the traditional export products that contributes strongly to the Gross Domestic Product [7]. According to the forecasts of the International Cocoa Organization, in the period 2021/2022, Ecuador is the third largest cocoa producer in the world [3]. The area planted with cocoa in 2020 was 590,579 hectares [4,8]. Approximately 70% of cocoa cultivation is concentrated in the hands of small producers, followed by 20% and 10% in the hands of medium and large producers, respectively [9]. In Ecuador there are two varieties of cocoa: Nacional (Arriba) and CCN51 that are cultivated in the coastal region and in the Ecuadorian Amazon. The "Arriba" cocoa is the emblematic variety, recognized worldwide by the confectionery industry as a fine-grade cocoa produced in Ecuador, with fruity and floral aroma and flavors [10–12]. The CCN51 variety is characterized by its fat content, productive capacity, and disease tolerance [5,13].

In the provinces of Orellana and Sucumbíos in the Ecuadorian Amazon, the first cocoa plantations were carried out with genetic material from the Nacional and Trinitario groups [14]. In 2005, the PRONORTE program, through a selection process of mother plants, identified eight cocoa ecotypes with high tolerance to diseases and with productive characteristics that exceeded the yield per hectare of Arriba and Trinitario cocoa. The selected mother plants were named "Super árbol" (Arriba × Trinitario) and in some cases they are promoted with the names "Sacha Gold" and "Brown Gold" cocoa, but this genetic material does not enjoy intellectual property because its selection did not come from an experimentally designed and directed hybridization process. Since 2010, the "Super árbol" cocoa was massively multiplied by different nurseries and was disseminated by local government programs so that in 2018 it constituted 47% of the cocoa area implemented in the province of Orellana [15,16]. The knowledge about "Super árbol" cocoa is mostly empirical; therefore, the generation of scientific information is necessary to define its characteristics.

The genetic variety, ripeness, fermentation, drying, and storage of cocoa influence the processing and specific characteristics of new specialty chocolate concepts [8]. However, fermentation is a key process because it produces biochemical reactions that give rise to aroma and flavor precursors, decrease acidity, astringency, bitter taste, and facilitate drying [9,10]. Poor control in the fermentation process will result in grains with undesirable characteristics for the industry, with underdeveloped aroma and flavor profiles [9,11].

This variation influences the physical and chemical quality of cocoa and is due to fermentation times and the type of fermenter (wooden crates, jute sack, heap, wooden box, and plastic tub) [11,12]. On the other hand, pre-drying is a technique that allows the development of a greater aromatic intensity of cocoa, and floral, fruity, and nutty flavors. In this sense, the present study aimed to evaluate the different fermentation techniques in cocoa (*Theobroma cacao* L.) "Super árbol" and "Arriba" by analyzing the physical and chemical characteristics in the climatic conditions of the Joya de Los Sachas canton, province of Orellana to establish the characteristics of "Super árbol" cocoa with respect to the "Arriba" variety that constitutes the reference variety in Ecuador.

## 2. Materials and Methods

### 2.1. Study Area

The experiments were carried out in the Ecuadorian Amazon, in the Joya de los Sacha canton, province of Orellana, at the Central Experimental Station of the Amazon (EECA) of the National Institute of Agricultural Research (INIAP) and the company "CacaoExport" (Figure S1). This area has a monthly rainfall of 275.64 mm, relative humidity of 80.6%, an average temperature of 25 °C [17,18], and a population of 39,352 inhabitants, according

to the projection for 2020 [4]. In this sector of Ecuador, since the 1970s, the villagers have been growing coffee and cocoa. The farmers own about 300,000 m$^2$ of land, where they have multiple uses for the land [17–19]. The Autonomous Decentralized Government of the Province of Orellana (GADPO) from 2011 to 2018 established 14.89 km$^2$ of cocoa, of which 9.35 km$^2$ was planted with "Super árbol" cocoa, and this had an impact on 1506 families [18,20].

### 2.2. Design of the Experiment

The "Super árbol" cocoa was obtained from CacaoExport. The "Arriba" cocoa was obtained from the research trials of the EECA of INIAP. The cocoa pods were harvested at physiological maturity. Considering that 2 types of fermenters (wooden box and jute bag) and 2 fermentation methods (with pre-drying and with pulp) were used for each type of cocoa, the resulting treatments were eight as indicated in Table 1. For each treatment, 45 kg of cocoa beans were used as fermentation mass. Three replicates were carried out for each treatment, at a single harvest time, May 2021. The experimental unit of the treatments consisted of 2 kg of cocoa beans that were placed in cloth nets in the center of the fermentation mass. A total of 315 kg of cocoa with pulp of each type of cocoa was used.

**Table 1.** Treatments performed in the research.

| Identity | Treatment |
| --- | --- |
| T1 | Arriba + Fermentation in jute sack + pre-drying |
| T2 | Arriba + Fermentation in jute sack + beans with pulp |
| T3 | Arriba + Fermentation in wooden box + pre-drying |
| T4 | Arriba + Fermentation in wooden box + beans with pulp |
| T5 | Super árbol + Fermentation in jute sack + pre-drying |
| T6 | Super árbol + Fermentation in jute sack + beans with pulp |
| T7 | Super árbol + Fermentation in wooden box + pre-drying |
| T8 | Super árbol + Fermentation in wooden box + beans with pulp |

Pre-drying consisted of shelling the cob 24 h after harvesting and draining, which consisted of placing a layer of 5 to 7 cm of almonds in a greenhouse-type solar dryer. It was removed every 30 min for the first 8 h, then left to stand for 16 h. The wooden boxes used had the following dimensions: 50 cm × 50 cm × 50 cm (length, width, height). Both the boxes and the bags were covered with banana leaves and polypropylene bags to insulate the heat. The fermentation process lasted 5 days, with a first removal at 48 h and periodic removals every 24 h.

At the end of fermentation, the cocoa beans were dried in a greenhouse-type solar dryer to a humidity of 7%. The physical and chemical analysis of the dried almonds was carried out in the Food Quality Laboratory of the EECA of INIAP. Figure 1 shows images of the types of cocoa, the fermenters, and the type of drying used in the research.

### 2.3. Measures Evaluated in the Experiment

#### 2.3.1. Physical Variables

The physical variables analyzed were fermentation rate, the content of beans shell, and weight of 100 beans. The fermentation rate was determined using the cutting test with a Magra 12 guillotine, model Teserba, B-Matthaei [20]. The cutting test consisted of randomly taking 100 cocoa beans and cutting them lengthwise by the guillotine. The cut beans were placed on a white base and classified as well-fermented almonds (brown to reddish-brown cotyledons throughout, and deep fermentation streaks), moderately fermented cocoa beans (cotyledons 50% brown in structure), violet cocoa beans (deep violet cotyledons), and slate-colored cocoa beans (blackish gray cotyledons with a compact appearance) [20,21].

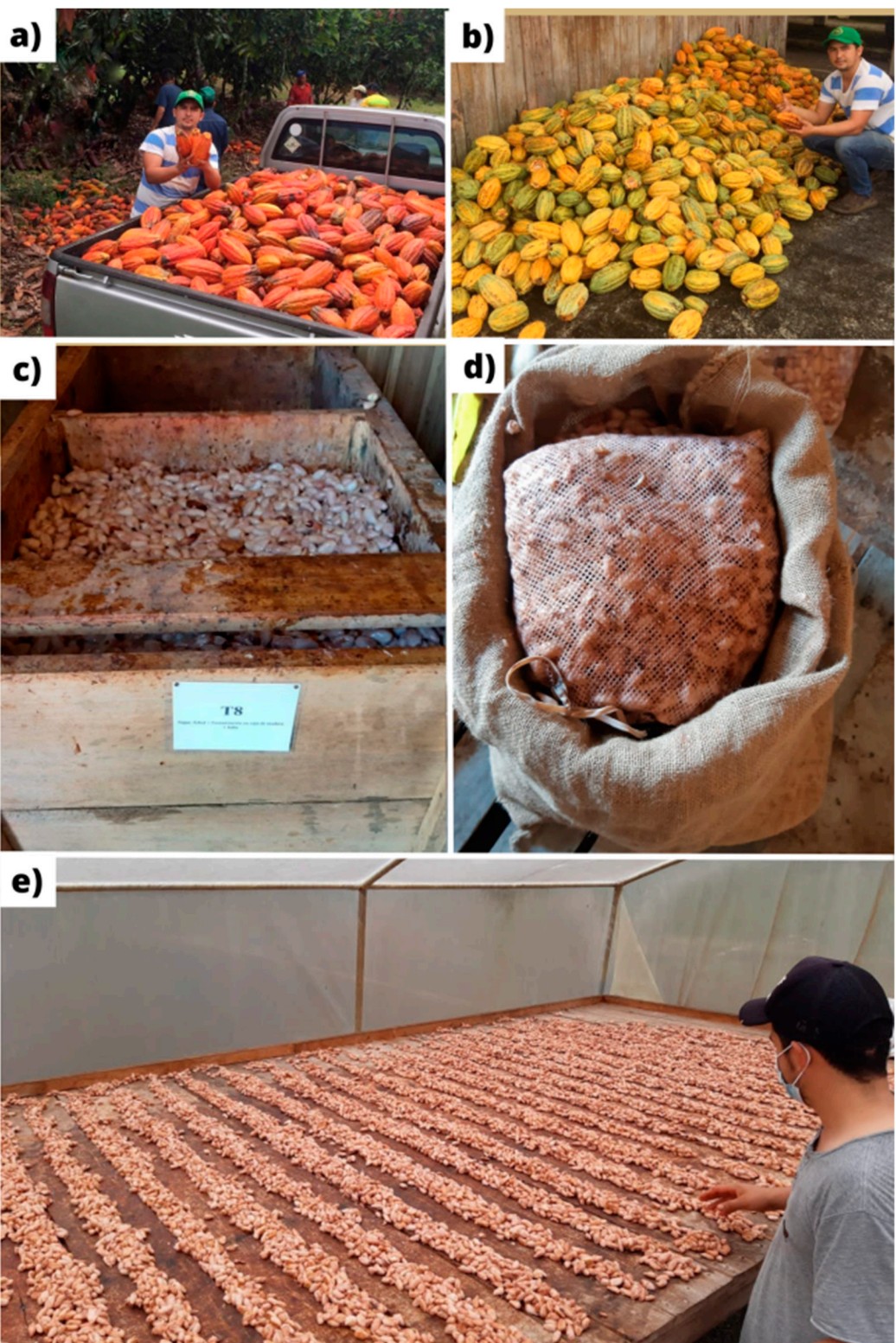

**Figure 1.** (**a**) Cacao pods "Super árbol"; (**b**) cacao pods "Arriba"; (**c**) wooden box; (**d**) jute sack; (**e**) solar drying-type greenhouse.

The weight of 100 cocoa beans and the percentage of beans shell were measured on a Citizen analytical balance, model CX 220. To obtain the percentage of beans shell, 30 g of almonds were weighed, the hulls and cotyledons were removed, each part was weighed, and the proportion of beans shell was calculated [20–22].

### 2.3.2. Chemical Variables

The chemical variables analyzed were pH, ash content, lipid, protein, and polyphenols. The pH of the cotyledon was evaluated in a 1:10 solution of cocoa powder in distilled water with a potentiometer [21]. The ash content was determined by gravimetry, incinerating 1 to 2 g of the dry sample at 500 °C for 4 h. The organic matter was oxidized and the resulting ash was considered the mineral part of the analyzed sample [23,24].

The lipids were extracted from the cocoa powder with petroleum ether by continuous Soxhlet extraction for eight hours, the solvent was recovered, and the containers were dried in an oven for two hours. The percentage of fat was determined as a function of the portion of fat extracted [25]. Protein content was determined from 0.5 g of the dehydrated sample by the Kjeldahl method. The digestion was carried out with 10 mL of concentrated sulfuric acid at 400 °C. After distillation, it was titrated with 0.3 N sulfuric acid until the color changed of the mixed indicator. The total polyphenols were extracted from the degreased cocoa powder with a 70% methanol aqueous solution by continuous magnetic stirring for 45 min (Figure S3). The extract was filtered and an aliquot was taken for colorimetric reaction with Folin Ciocalteu's reagent. For quantification, a Perkin Elmer UV-VIS Spectrophotometer, model Lambda 25 was used at a wavelength of 760 nm (Figure S2); total polyphenols are reported as mgGAE/g DW [25,26].

### 2.4. Statistical Analysis

Statistical analysis was performed in the free package R and RStudio, obtained through the libraries "corrplot", "PerformanceAnalytics", "car", "ggpurb". In addition, the statistical analysis software Infostat version 2020 was used to generate the analysis of variance and the Tukey test.

The study was conducted under a completely randomized design with three factor or independent variables. The dependent variables totaled eight, as indicated in Table 2.

**Table 2.** Independent and dependent variables for the experimental design.

| Independent Variables | | Dependent Variables |
|---|---|---|
| **Variables** | **Levels** | |
| Genetic Material (G) | g0 = "Arriba" | Fermentation rate (%) |
| | g1 = "Super árbol" | Weight of 100 beans (g) |
| Fermenter Type (F) | f0 = Jute sack. | pH cotyledon |
| | f1 = Wooden box | Beans shell (%wt) |
| Fermentation Method (MF) | mf0 = Pre-drying | Protein (%wt) |
| | mf1 = Beans with pulp | Ash (%wt) |
| | | Lipid (%wt) |
| | | Total polyphenols (mgGAE/gDW) |

The independent variables that intervened in the experiment had two levels. Therefore, the general model corresponds to a three-factor model with three replications, as detailed below.

$$y_{ijkl} = \mu + \tau_i + \beta_j + \gamma_k + \left(\tau\beta_{ij}\right) + (\tau\gamma_{ik}) + \left(\beta\gamma_{jk}\right) + \left(\tau\beta\gamma_{ijk}\right) + \epsilon_{ijkl}$$

where:

$$\begin{cases} \mu, \text{ is the total average} \\ \tau, \text{ is the genetic material factor} \\ \beta, \text{ is the fermenter type factor} \\ \gamma, \text{ is the fermentation method factor} \\ \varepsilon \text{ represent the errors} \end{cases}$$

The hypotheses of the study were verified by the results of the analysis of variance. Both the null and alternative hypotheses of the treatments and their interaction are described below:

Treatment effect hypotheses

$$\begin{cases} H_0 : \tau_i = \beta_j = \gamma_k = 0 \\ H_1 : at\ least\ one\ is\ nonzero \end{cases}$$

Hypotheses on treatment interactions

$$\begin{cases} H_0 : (\tau\beta_{ij}) = (\tau\gamma_{ik}) = (\beta\gamma_{jk}) = 0 \\ H_1 : At\ least\ one\ interaction\ is\ different\ from\ zero \end{cases}$$

In the execution of the experiment, data for the eight dependent variables evaluated were collected in the eight treatments applied. The number of observations in each treatment totaled 24. Due to the nature of the variables and the number of observations, a Spearman correlation analysis was performed to determine the possible relationships and associations between the observed variables. This allowed optimizing the execution of the ANOVA models.

An ANOVA model was identified for each of the four selected variables. Each model was estimated and validated. The least-squares method was used for estimation because ANOVA models are part of linear models. The F-test was used to validate the estimators. Homogeneity of variance was analyzed using Levene's test and to establish the difference between means of the treatments, Tukey's test was used with a confidence level of $p$-value $\leq 0.05$.

## 3. Results

Figure 2 indicates the Spearman correlation between the observed dependent variables. This analysis identified and prioritized four variables to apply the analysis of variance: bean shell, fermentation rate, total polyphenols, and lipid. The other dependent variables were discarded because the experiment was completely estimated with the selected variables (Table S1).

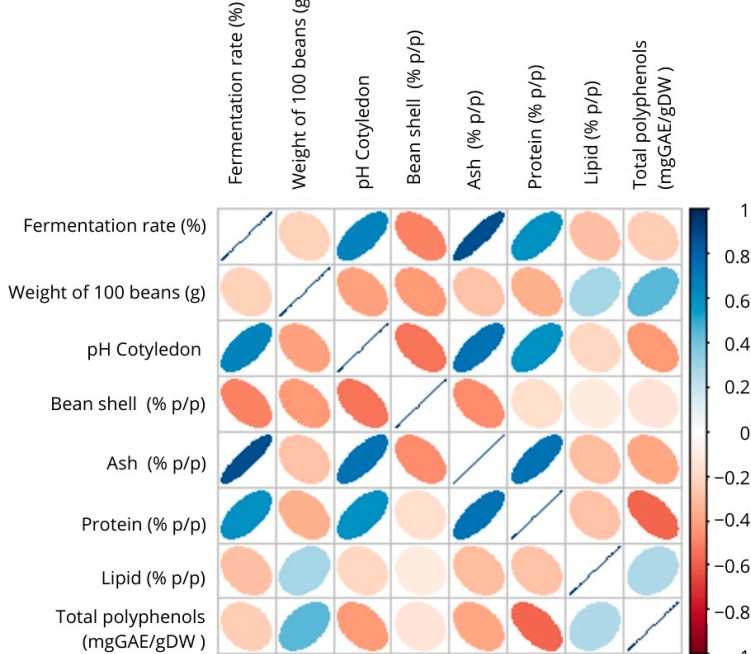

**Figure 2.** Spearman correlation analysis between the observed variables.

The results obtained from the analysis of variance are summarized in Table 3. This table indicates the models established for each dependent variable, the level of significance of the study factors. as well as the interactions for each of the dependent variables determined in the correlation analysis. The independent variables considered in the experiment did not statistically intervene in the percentage of lipid in the cocoa analyzed.

**Table 3.** Summary of the analysis of variance for the prioritized variables.

| Variable | Components | Coefficients | Degrees of Freedom | Means Squares | F-Value |
|---|---|---|---|---|---|
| Fermentation rate (%) | Intercept | 62.96 | | | |
| | G | 10.50 | 1 | 234.375 | 50.153 * |
| | F | 0.67 | 1 | 77.042 | 16.486 * |
| | MF | 22.42 | 1 | 3015.042 | 645.171 * |
| | G × F | −8.50 | 1 | 108.375 | 23.191 * |
| | Residuals | | 19 | 4.673 | |
| Been shell (% p/p) | Intercept | 13.88 | | | |
| | G | 3.89 | 1 | 14.570 | 27.366 * |
| | F | 0.71 | 1 | 5.743 | 10.786 * |
| | MF | 2.45 | 1 | 35.624 | 66.910 * |
| | G × F | −0.40 | 1 | 6.720 | 12.622 * |
| | G × MF | −6.79 | 1 | 27.392 | 51.448 * |
| | F × MF | −5.49 | 1 | 13.321 | 25.019 * |
| | G × F × MF | 5.03 | 1 | 9.475 | 17.797 * |
| | Residuals | | 16 | 0.532 | |
| Polyphenols (mg/g) | Intercept | 62.63 | | | |
| | G | −10.70 | 1 | 736.798 | 19.985 * |
| | F | 11.16 | 1 | 195.785 | 5.311 * |
| | MF | −10.19 | 1 | 157.737 | 4.279 ns |
| | G × F | −10.89 | 1 | 177.932 | 4.826 * |
| | G × MF | 10.13 | 1 | 154.027 | 4.178 ns |
| | Residuals | | 18 | 36.867 | |
| Lipid (% p/p) | Intercept | 47.68 | | | |
| | G | −3.31 | 1 | 16.033 | 2.749 ns |
| | F | −1.38 | 1 | 0.453 | 0.078 ns |
| | MF | −3.71 | 1 | 13.249 | 2.271 ns |
| | G × F | 2.18 | 1 | 0.003 | 0.001 ns |
| | G × MF | 3.30 | 1 | 2.043 | 0.350 ns |
| | F × MF | 3.27 | 1 | 1.935 | 0.332 ns |
| | G × F × MF | −4.27 | 1 | 6.844 | 1.173 ns |
| | Residuals | | 16 | 5.833 | |

*: statistically significant difference $p = 0.05$; ns: no significant differences.

Figure 3 shows the differences between the means of the dependent variables in the estimated models. When the categories or levels of the independent variables are farther apart, the significant difference between their means will be greater.

### 3.1. Physical Variables

In the fermentation percentage, the most relevant variance factor was the fermentation method, followed by the genetic material and type of fermenter. The "Super árbol" cocoa obtained results between 64.3% and 95%, while the "Arriba" cocoa obtained fermentation percentages between 63.0% and 85.7%. The "Super árbol" cocoa achieved the highest percentage of fermentation in the wooden box with the pre-drying method, while "Arriba" cocoa achieved the best fermentation percentage with pre-drying, regardless of the use of a wooden box or jute sack.

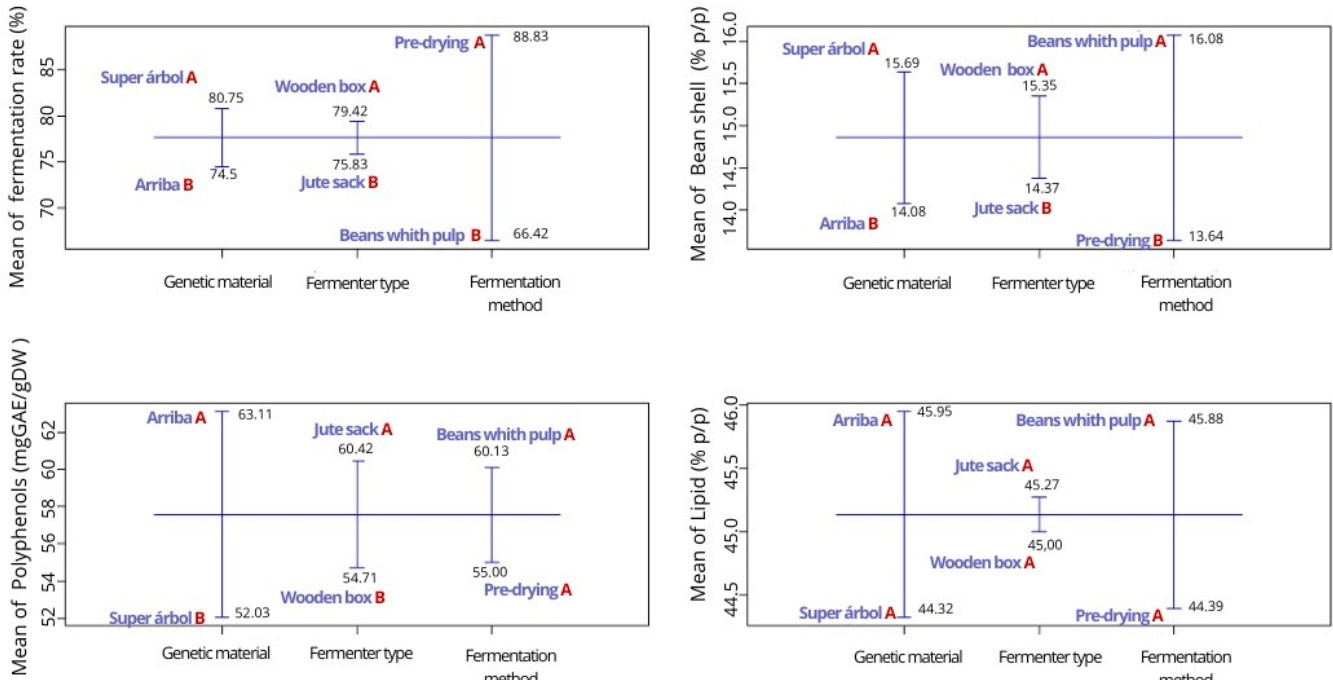

**Figure 3.** Relationship between factors of the experiment and the physicochemical variables analyzed. (A,B): Capital letters indicate significant differences between levels of the independent variables ($p < 0.05$).

In the percentage of bean shell, the most influential factor was the fermentation method, followed by the genetic material and type of fermenter. In the case of cocoa fermented with pulp, the results were 16.08%, while with pre-dried cocoa the percentages were reported as low as 13.64%. The "Super árbol" cocoa reported values between 13.3% and 17.8% while the "Arriba" cocoa between 11.5% and 16.3%. In this variable, both "Super árbol" and "Arriba" cocoa had lower values when the cocoa beans were subjected to pre-drying and fermentation in a jute bag.

### 3.2. Chemical Variables

For the fat percentage variable, no factor showed significant differences, so it can be considered that the amount of fat in the "Super árbol" cocoa is equivalent to that of the "Arriba" cocoa; therefore, it is not possible to establish an advantage between the genetic materials, fermentation method, or type of fermenter with respect to this variable.

In total polyphenols, the factor with the greatest influence of variation was the genetic material. The "Arriba" cocoa reported results between 57.23 and 79.18 mgGAE/gDW, while the "Super árbol" cocoa reported results between 48.46 and 55.54 mgGAE/gDW. The results are consistent with those reported by Nazaruddin et al., 2001, which indicated that total polyphenols for cocoa beans ranged between 34 and 60 mgGAE/gDW [26]. The high concentration of polyphenolic compounds contributes to the bitter and astringent taste of unroasted cocoa and can be attributed to the generation of the brown color achieved in fermentation through the role of polyphenol oxidase in enzymatic browning. Polyphenols are reduced during roasting by polymerization and/or oxidation and, consequently, perceived bitterness and astringency decrease [27]. Polyphenols in chocolate and cocoa products may differ according to country of origin and preparation methods [28].

## 4. Discussion
### 4.1. Physical Variables

Fermentation is a fundamental step in obtaining cocoa beans of optimum quality because it allows the production of chocolate with the flavor and aroma required by the

market. The purposes of fermentation are: draining of mucilage, death of the embryo due to the introduction of acid to the cotyledon and the high temperature, development of aroma and flavor precursors, change of color of the kernel, and reduction of bitter taste and astringency [29,30].

Figure 3 shows the fermentation percentage was positively influenced by the pre-drying method and the wooden box as a fermenter. In addition, the "Super tree" cocoa had a higher fermentation percentage than "Arriba". The reasons can be attributed to the fact that all factors favor the drainage of exudates. Regarding the genetic material, it should be considered that the "Super árbol" cocoa beans are smaller and lighter than the "Arriba" cocoa beans, while the specific surface area of the "Super árbol" beans are larger. These results constitute a favorable characteristic for the post-harvest process of "Super Árbol" cocoa, which was selected a priori only for its apparent yield characteristics and its tolerance to pests and diseases. The increase in fermentation significantly influenced by pre-drying is consistent with the study of ref. [31], which indicated that the fermentation index of cocoa, used to control the degree of fermentation increased with the passage of days of storage of cocoa in pods, being a more evident behavior in the 3 days of fermentation because the condensation product from this time was less soluble.

According to NTE INEN 176:2022-06, Ecuadorian cocoa beans are classified by quality from grade 1 to 3, with 1 being the highest quality and 3 the lowest quality. One of the requirements in the quality standard is the fermentation percentage, which ranges from 53% for grade 3 to 75% for grade 1. The fermentation percentage obtained with the different treatments applied in this study meets this requirement [32]. An adequate level of fermentation makes it possible to guarantee the series of biochemical transformations that give rise to aroma and flavor precursors. Among the biochemical transformations that occur during fermentation is the elimination by the exudate of pigment cells, these cells contain polyphenols of the an-tocyanin type that change the color of the almond from purple to brown [11]. Epicatechin and catechin are oxidized to quinones and condensation of proteins and polyphenols occurs, decreasing astringency. Methylxanthines are lost by 20%, resulting in a decrease in bitter taste. All these compounds affected during fermentation, including volatile acidity (acetic acid) and the volatile fraction, cause an increase in the organoleptic quality of cocoa, resulting in floral and fruity notes [3,30,33]. Undoubtedly, the post-harvest stage that most influences chocolate quality is bean fermentation [34,35].

Similar results to those reported in this study were found in the work carried out by ref. [36] in the province of Manabí, Ecuador. In this work, they analyzed the effect of fermentation time with the type of fermenter (wooden crates, jute bag, heap, and plastic tub) on physical and chemical quality of Nacional type cocoa (above). The results showed that the fermentation time influenced the physical quality of the beans, reaching 75% of brown beans after five days of fermentation and thus a decrease in the violet color inside the beans. Likewise. ref. [37] reached around 80% of fermented beans in the different National cocoa clones (Arriba) EET103, EET559, EET576, and EET577 using wooden crates in the province of Guayas, Ecuador.

In the percentage of bean shell, also known as hulls or husks, the factor with the greatest influence was the fermentation method, followed by the genetic material and type of fermenter. The average for "Super Arbol" cocoa was 15.69 while for "Arriba" it was 14.08. These percentages are consistent with the range of 12% to 20% considered to estimate the annual production of these agro-food coproducts, which reaches approximately 600 thousand tons [38]. Cocoa bean shells are disposed of as waste and underutilized as fuel for boilers, animal feed, fertilizer, and occasionally as an ingredient in local biscuit production and up to 5% in chocolate production [5,6,38]. Additionally, it is possible to consider this material as a biomass resource to eliminate pollutants [39].

Similar results of percentage of cocoa shell were presented by ref. [40], who reported values between 13.78% and 14.16% in cocoa fermented in wood and plastic boxes, while ref. [22] reported values of 18.92%. The results of this study are among the lowest percent-

ages when compared with the aforementioned works, which is a positive characteristic for the treatments applied.

"Super tree" cocoa produces a higher amount of bean shell compared with "Arriba" cocoa, which could be an unfavorable characteristic for yield in the chocolate manufacturing industry [41]. However, there are studies such as those in refs. [42–44] that identify bean shell cocoa as a valued coproduct for the food industry due to its content of bioactive compounds such as polyphenols, flavanols, antioxidants, pectin, fibers, and methylxanthines [6].

### 4.2. Chemical Variables

Fat percentage did not vary significantly among the independent variables studied. The "Super árbol" cocoa reported values between 43.76% and 45.15% while the "Arriba" cocoa had values between 43.97% and 47.68%. Similar values were obtained [45] in a study of "Arriba" cocoa from the Ecuadorian Coast and Amazon. The values reported were $45.61 \pm 1.27$ to $52.13 \pm 0.58$ g/100 g PS with the particularity that cocoa from the provinces and cantons of the Amazon region had higher fat content than cocoa from the Coast. In the work carried out in ref. [4] it was found that CCN51 cocoa had 51.81% fat while "Arriba" cocoa had 50.81%. The meta-analysis in ref. [46] mentioned that the range of fat in cocoa beans is between 50% and 57%. This lipid is responsible for chocolate melting and aroma. According to ref. [22], forastero-type cocoa beans have fat contents higher than 52% while fine cocoa beans report values lower than 50%. Fat content is influenced by geographic location and genetic variation, but not by fermentation methods.

The results in this study showed that polyphenols decrease when using the wood box fermenter due to the genetic material. In addition, there was a decrease when pre-drying is applied, although it did not show a significant difference. This is consistent with those reported by Nazaruddin et al., 2001, who indicated that total polyphenols for cocoa beans ranged from 34 to 60 mg GAE/g DW [22,47]. Ecuadorian "Arriba" cocoa harvested in six different production areas have reported values between 34.67 and 100.05 mgGAE/g DW. Cocoa harvested in the cantons of the Amazon region of Ecuador have presented results between 42.75 and 71.66 mg GAE/g DW, while samples collected in the littoral region presented polyphenols between 33.55 and 47.40 mg GAE/g DW [48]. These values are similar to those obtained in this study for "Arriba" cocoa but higher for "Super árbol" cocoa.

The high concentration of polyphenolic compounds contributes to the bitter taste and astringency of unroasted cocoa. Polyphenols are reduced during roasting by polymerization and/or oxidation and, consequently, the perceived bitterness and astringency decreases [49]. Polyphenols in chocolate and cocoa products may differ according to country of origin and preparation methods [50]. For example, Nazaruddin et al., 2006, indicated that pod storage and pulp preconditioning (pressing) caused a significant reduction in the content of polyphenol compounds. However, the effect on xanthine alkaloid content was not significant, so it was suggested that the reduction of polyphenols during fermentation was mainly due to exudation [31].

Polyphenol reduction is an important parameter during fermentation because it is related to astringency (unpleasant taste of the bean) and bitter taste before the manufacture of chocolates and, in post-harvest processes, cannot be eliminated. However, in recent years polyphenols have gained prominence due to their antioxidant action and beneficial effects on health. Therefore, a certain reduction in the level is required to achieve cocoa beans with good flavor, but a balance is sought so as not to lose the functional characteristics (antioxidative activity) [51]. There is a proposal that a content of 50 mg GA/g allows a balance between flavor and functional characteristics (antioxidant activity) [36]. Therefore, according to the results reported in the research, T8 balanced chocolate would be obtained in terms of flavor without astringency and polyphenol content that allows the contribution of the antioxidant activity attributed to dark chocolate.

The results obtained in this research suggest that it is necessary to perform analyses to determine the polyphenolic and fatty acid profile of the beans and the "Super árbol"

cocoa bean shell, so that it can be inferred whether these materials meet the characteristics of functional foods that contribute to a circular economy of the Amazonian peoples of Ecuador. Choosing cocoa with attributes that will produce premium-quality chocolate is important to the industry. However, farmers sell their product at a low price because the cocoa value chain is disorganized. Farmers have to sell their produce through a series of agents and traders, who pay only according to the weight of the dried beans; therefore, cocoa drying is carried out in areas open to sunlight, and the beans are prone to mold and other bacterial problems, leading to low quality cocoa. Consequently, investment in cocoa production and post-harvest is very risky for smallholder farmers [45,52].

## 5. Conclusions

The "Super árbol" cocoa reported fermentation percentages between 64.33 and 95%, bean shell percentages between 13.28% and 18.08%, and polyphenol content between 48.46 and 55.54 mg/g, which gives it favorable quality characteristics for the chocolate industry. This genetic material would have a higher fermentation percentage and lower polyphenol content than the "Arriba" cocoa. However, by having a higher bean shell percentage, the "Super árbol" cocoa will produce more coproduct than the "Arriba" cocoa, but this disadvantage can be mitigated with the application of fermentation methods such as pre-drying, which have a significant influence on this variable when seeking to use it as a fuel or functional food.

In the variable's fermentation percentage, bean shell percentage, and total polyphenol content, the pre-drying method is a factor that positively influences the "Arriba" and "Súper árbol" cocoa, so local producers could use it as an alternative to improve the quality of cocoa and to counteract the climatic conditions of the Joya de los Sachas, which are characterized by high rainfall and high percentages of relative humidity. The effect of the type of fermenter in this research was not relevant. However, it is important to point out that the turning work is strenuous when jute bags are used and requires at least two people to handle one bag, so using this type of fermenter has the disadvantage of requiring more personnel.

For Super árbol and Arriba cocoa, the fermentation method that had a positive effect on the variables analyzed was pre-drying. The type of fermenter with the most favorable effect was the wooden box. Consequently, the application of T3 ("Arriba", wooden box, predrying) and T8 ("Super árbol", wooden box, predrying) is suggested to local producers because it would allow them to obtain cocoa with fermentation percentages within the national standards, bean shell percentages acceptable for the industry, and total polyphenol content that allows a balance between a low astringency flavor and antioxidant activity.

The main limitations in the research process were the delay in data collection due to the presence of adverse weather conditions in the Amazonian area of Ecuador, which lengthened the time for sampling and data analysis. In addition, since the research was carried out in a worldwide pandemic situation of COVID-19, there were problems in the mobilization and entry to farms in the study area. Finally, it is important to focus efforts on new research to learn more about the characteristics of cocoa varieties in the Amazon region and correlate them with resistance to pests and diseases. All this will serve to provide important information from academia and research centers to local farmers and make their agro-industrial processes more sustainable and profitable.

**Supplementary Materials:** The following supporting information can be downloaded at: https://www.mdpi.com/article/10.3390/su14137564/s1, Figure S1. Study area; (a) Ecuador; (b) Orellana, La Joya de los Sachas; (c) Planting areas; Figure S2. Calibration curve for total polyphenols; Figure S3. Total polyphenol analysis photograph; Table S1. Effect of the treatments on the dependent variables analyzed.

**Author Contributions:** Conceptualization, M.S.-C., S.V.-S., A.B.-C., M.A.-D., T.V.-T., S.S.-C. and C.M.-R.; methodology, M.S.-C., S.V.-S., A.B.-C., M.A.-D., T.V.-T., S.S.-C. and C.M.-R.; software, M.S.-C., S.V.-S., A.B.-C., M.A.-D., T.V.-T., S.S.-C. and C.M.-R.; formal analysis, M.S.-C., S.V.-S., A.B.-C., M.A.-D., T.V.-T., S.S.-C. and C.M.-R.; investigation, M.S.-C., S.V.-S., A.B.-C., M.A.-D., T.V.-T. and S.S.-C.; writing—original draft preparation, M.S.-C., S.V.-S., A.B.-C., M.A.-D., T.V.-T., S.S.-C. and C.M.-R.; writing—review and editing, M.S.-C. and C.M.-R.; project administration, M.S.-C., S.V.-S., A.B.-C., M.A.-D., T.V.-T., S.S.-C. and C.M.-R.; resources, M.S.-C., S.V.-S., A.B.-C., M.A.-D., T.V.-T. and S.S.-C. All authors have read and agreed to the published version of the manuscript.

**Funding:** The study was financed by the Escuela Superior Politécnica de Chimborazo, within the project "Dinamización de la economía mediante la implementación de un producto turístico de con-servación sostenible, para mejorar la calidad de vida y protección del ambiente en la comunidad Río Indillama, Parque Nacional Yasuni".

**Institutional Review Board Statement:** Ethical review and approval were waived for this study due to the fact that the study was carried out based on an informed consent and anonymity of the respondents.

**Informed Consent Statement:** Each respondent was informed in detail about the objectives of the study and how the data would be used. Each responded agreed verbally to participate in the study under an anonymity clause.

**Data Availability Statement:** All of the data supporting this study may be made available upon request to the first and second authors of the study.

**Acknowledgments:** The authors would like to thank the Instituto Nacional de Investigaciones Agropecuarias (INIAP) Estación Experimental Central de la Amazonía (EECA) and the Escuela Politécnica del Chimborazo (ESPOCH), Sede Orellana for having provided all the facilities to carry out this study.

**Conflicts of Interest:** The authors declare no conflict of interest.

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
