# Peer review of "New Characteristics in the Fermentation Process of Cocoa (Theobroma cacao L.) “Super Árbol” in La Joya de los Sachas, Ecuador"

_sustainability, doi:10.3390/su14137564_

Round 1

Reviewer 1 Report

In the manuscript, the authors described an interesting proposal to study the variations “Super tree” and “Ariba” cocoa (Theobroma cacao L.) in Ecuador. The proposal is adequate, the methodology is applicable, and the results are well-conducted. However, the authors demonstrated a certain neglection with the work, considering some mistakes in translation. In addition, the work must increase the quality of data to represent valuable information to the readers. The work presented here just confirms some proposes described in the literature, becoming a shallow work.

The Introduction is well-conducted and described all important topics to elucidate to the reader the issue, but the general fundament of the introduction is too long. I suggest to the authors summarize the introduction. In addition, I strongly suggest the authors reinforce why cocoa in Ecuador is so important, and why Ecuadorian chocolate (the main subproduct of cocoa) is important globally. The importance of the study is up to this moment still not clear. Such discussion must fundament the paper.

- On p. 2, l.50, please insert the country of the regions cited.

- On p. 2, l.57, please complete quotations of Ariba.

In the materials and methods section, the authors described sufficiently the methods. However, we observed the great neglect of the authors in does not revise the manuscript in some parts with no translation, or poor translates were observed. I strongly suggest to the authors revise the entire manuscript. In addition, the methodology must be well-referenced, considering the methods used were described in proper literature.

- On p. 3, l.105, 108, please convert ha to m².

- Figure 1 must be inserted in the supplementary material.

- In table 1, the authors used “domestic” to refer to “Ariba” kind of cocoa, as well as “super tree” and “super árbol”. The lack of standardization is still difficult reading Table 2(p.5), where different naming is evaluated. The authors must standardize the naming of plants and methods. I suggest “Ariba” and “Super tree” in the entire text.

- On p. 3, l. 119, the authors suggested four ways to ferment cocoa. Please cite a reference to evidence of the methodologies of fermentation.

- On p. 4 l. 140, please change “photographs” to “pictures”. In some parts of the text, the authors used some expressions not common in the English language, as well as literal translates. In addition, the authors in some parts of the text do not translate, e.g., p.5, l.158 “Valiables quimicas analizados”; p.5 l.179 “Tabla 2, and others (Figure 2, l.195, p.6; . Factores para el diseño del experimento”.  I strongly suggest English revisions to the entire text.

- On p.5 l.148, please cite a proper reference of the test described.

- On p.5 l.155, please cite the origin of the balance.

- On p. 5, l. 169, please change “ml” to “mL”. SI units are necessary.

- On p.5 l.174, please cite the origin, brand, and model of the UV-Spectrophotometer. The calibration curve of the analytical method must be available in supplementary information.

Results and discussion: In the results and discussion, the authors expressed some physical and chemical properties and compared them to the literature. The comparison is valuable and useful to establish some differences between the different variations of cocoa but does not present some news about the issue. In addition, the study must deliver a difference in comparison to other works.

- On p. 7, table 3, again some terms are not translated. Please fix it.

- On p. 8, figure 3 must be formatted, because the variables analyzed in the pictures are not in good resolution.

Conclusions are ok. Some elements to make different the paper from the literature is demanding. Supplementary information could demonstrate valuable information to the readers, but it was not attached to this paper.

References must be entirely revised in the form and content. The references must include names of species inadequate format, standardization of the name of the Journal and those abbreviations, name of the authors, and information.

Author Response

Dear reviewer, first of all we thank you for taking the time to thoroughly review our manuscript, this has helped us to improve the document in general. In that sense, we have made the corrections that respond to your comments and observations, which we describe below:

  • The translation of the work was revised in its entirety.
  • Complementary information was generated for the work, including the calibration curve used for the polyphenol analysis and the location map of the study.
  • The introduction was restructured and its content was compressed. This served to put into context the importance of cocoa in Ecuador and its main problems.
  • P. 1, 48-59 reinforced the importance of cocoa in Ecuador and Ecuadorian chocolate worldwide.
  • The requested mention of the countries in Q2, l.50 was made in Q1, l43 and 44.
  • The citation requested in Q2, l57 was not included, now it is included in Q2, l58.
  • The methodology references were entered as well as the translation revision.
  • The hectare units were changed to km2 , not changed to m2 as suggested because the numbering was visually too extensive.
  • Figure 1 was inserted in the supplementary material.
  • Standardization corrections of terms in Table 1 and throughout the document were made.
  •  P. 3, l. 119, now P3. L116 the expression four fermentation techniques was corrected since it was intended to refer to the resulting treatments.
  •  P. 4 l. 140, the word photographs was changed to images.
  •  P. 5 l.148, the requested reference was cited.
  •  P. 5 l.174, the brand and model of the spectrophotometer was detailed and in the complementary information the calibration curve for polyphenols was added.
  • The discussion of the work was enriched according to the observations.
  • The translation of Table 3 was corrected.
  • The resolution of Figure 3 was improved
  • The references were revised in their entirety
  • The conclusion was improved and complementary information was attached in the system.

Reviewer 2 Report

Comments

(Manuscript No. 1730032)

The manuscript titled “New characteristics in the fermentation process of cocoa (Theobroma cacao L) ""Super Árbol"" in La Joya de los Sachas, Ecuador” is well written and evaluate different fermentation techniques in the "Super Árbol" cocoa.

(1) What is the difference between "jute bag" and "jute sack", "wooden box" and "wooden crate" in Table 1? If not, we suggest that it be standardized.

(2) It is suggested to add an appropriate description of the meaning of the graphs in Figure 2.

(3) The numbering of 2.5. in the text is wrong. There is an error in the table title of Table 2. Please correct and check.

(4) Why tukey’s test and levene’s test were chosen for the article? Please explain the reason.

(5) Some new literatures might be help the authors to further deepen the understanding of reaction mechanism as well as newest developing in this field (Bioresource Technology, 2021, 332: 125086   Study on the Hg0 removal characteristics and synergistic mechanism of iron-based modified biochar doped with multiple metals).

Author Response

Dear reviewer, first of all we thank you for taking the time to thoroughly review our manuscript, this has helped us to improve the document in general. In that sense, we have made the corrections that respond to your comments and observations, which we describe below:

(1) What is the difference between "jute bag" and "jute sack", "wooden box" and "wooden crate" in Table 1? If not, we suggest that it be standardized.
Indeed as you suggest it was a standardization problem in the document which was corrected.

(2) We suggest adding an appropriate description of the meaning of the graphics in Figure 2. 
The description of the images in Figure 2 were modified by specifying what each one indicates.

(3) The numbering of 2.5. in the text is wrong. There is an error in the title of the table in Table 2. Correct and verify it. 
The numbering was corrected.

(4) Why were Tukey's and Levene's tests chosen for the article? Explain why. 
Levene's test was used to test the homoscedasticity of the model, i.e. that the errors have a normal distribution. Tukey's test was used in conjunction with an ANOVA analysis, it served to evidence all differences in means according to treatments.

(5) Some new publications could help the authors to further deepen the understanding of the reaction mechanism as well as the most recent development in this field (Bioresource Technology, 2021, 332: 125086 Study on Hg 0 removal characteristics and synergistic mechanism of iron-based compounds). modified biochar doped with multiple metals). 

Finally, the recommended reading was of great help in understanding the alternatives for the use of waste from chocolate production, it served to argue the discussion of the results obtained on the percentage of husk.

Reviewer 3 Report

Good day! Thank you!

Author Response

Dear Reviewer, we appreciate your contribution to our work, this helps us to continue improving in the world of research.

Reviewer 4 Report

Dear Corresponding Author

I checked your paper. Unfortunately, it is not qualified enough to publish in Sustainability journal. Your methods are weak and presentation of your analysis is not well organized. There are so many spelling errors and grammatical problems and even errors in writing the numbers.

After your extensive revision about your paper I think you may submit it again.

Regards

Reviewer

Author Response

We thank you for your comments, the document has been improved thanks to the comments of all the reviewers, we hope you like this new version.

Thank you, good day.

Reviewer 5 Report

Thank you for the opportunity to review on an interesting topic „New characteristics in the fermentation process of cocoa (Theobroma cacao L) “Super Árbol” in La Joya de los Sachas, Ecuador“.

Major concerns

It seems that the Methods Unit should be based on clearer subsections (e.g. Study Area and Design (1), Measures (2) and Statistical Analysis (3)).

In the Methods Unit Authors are required to determine dependent and independent variables or to identify the main focus of this study or what the Authors are trying to assess (the dependent variable), as well as the variable (s) the Authors will manipulate in order to cause an effect (the independent variable(s)).

L 204: Results and discussion section must be separated into two separate subsections such as „Results“ and „Discussion“.

It seems necessary to adjust Table 3 (content and English).

Figure 3 presents the results of the ANOVA test. It seems necessary to present this figure in the table. I am not convinced whether ANOVA test results mean correlation (as the Authors point out). Rather, this test would confirm the statistical significance of the differences between the variables being analysed.

Results must be described according to the selected design of the experiment. For example, it can be recommended to assess change or in order to determine whether the manipulation of independent variable has affected dependent variable. The most effective way to do this is through pre- and post-testing.

It seems that the manuscript does not include a paragraph of Limitations.

The Authors referred to the Conclusions of the study as follows: „In this sense, the present study aimed to evaluate the different fermentation techniques in cocoa (Theobroma cacao L.) "Super tree" and "Arriba" by analyzing the physical and chemical characteristics in the climatic conditions of the Joya de Los Sachas canton, province of Orellana to establish the characteristics of "Super Tree" cocoa with respect to the "Arriba" variety that constitutes the reference variety in Ecuador“. Clear and concrete conclusions must be drawn on the basis of these research objectives and the main results. In addition, there seems to be a lack of practical implications.

Minor concerns

What is the number of Ethical approval issue?

References must be written in accordance with the recommendations of MDPI.

Author Response

Dear reviewer, first of all we thank you for taking the time to thoroughly review our manuscript, this has helped us to improve the document in general. In that sense, we have made the corrections that respond to your comments and observations, which we describe below:

  • Materials and Methods was adjusted to the suggested organization.
    In the Methods Unit, the Authors should determine the dependent and independent variables or identify the main focus of this study or what the
  • Authors are trying to evaluate (the dependent variable), as well as the variable(s) that the Authors will manipulate to cause an effect (the independent variable(s)).
  • In section 2.5.    Statistical Analysis, the independent and dependent variables were indicated in the
  • The separation of results and discussion was carried out as suggested in section 2.5.
  • The translation of the information in Table 3 was revised as in the whole document.
  • The correlation information of the treatments was presented in Figure 3 because it allows visualizing the interaction of the independent variables (genotype, type of fermenter and fermentation method) for each dependent variable that showed significance in the correlation analysis.
    For each dependent variable, the influence of the independent variables P8.l233 - 262 was punctuated.
  • The main limitations were included.

Minor concerns

  • The ethics statement was included and all references were reviewed and adjusted to the MDPI format.

Round 2

Reviewer 1 Report

The authors still demonstrated neglection with the work, considering some mistakes in translation (l.4, p.135-139). The work still must increase the quality of data to represent valuable information to the readers, and still confirms descriptions in the literature. Results and discussion are poor. References must be revised. I still suggest the paper is not suitable for publication in the Sustainability journal in this version.

-Errors of translation are still observed (l.4, p. 135-139)

-The supplementary information presented just a calibration curve, and presented translation mistakes.

-The introduction increases the quality.

-Figure 1 cannot be observed in the manuscript, just in the supplementary information.

-The discussion is still poor.

-References must be revised in the format.

-Conclusion is still poor.

Author Response

Dear reviewer, we thank you for your comments and apologize for the oversights in the document. Each of the corrected errors allows us to improve. Your comments have been very strong and we believe they have contributed a lot to improve our manuscript. We hope you like this new version and look forward to continuing to process the manuscript.

Point 1: The authors continue to show negligence with the work, considering some translation errors (l.4, p.135-139). 
Response 1: Thank you for your comment. The document was revised and the requested translation was made.

Point 2: The supplementary information presented only one calibration curve and had translation errors.
Response 2: The translation was corrected and supplementary information from laboratory work and extra data were added.

Point 3: The introduction increases the quality.
Response 3: We appreciate the comments, even though we have noticed that the evaluation in this section has decreased, we complied with the requirements of you and other reviewers in the first round, we hope you like it now. Thank you.

Point 4: Figure 1 is not visible in the manuscript, only in the supplementary information.
Response 4: Figure 1 has been removed from the paper and is only in the supplementary information, as you suggested.

Point 5: The discussion is still poor.
Response 5: Some aspects were added to the discussion, they are distinguishable because they have orange coloring.
Point 6: The references should be revised in the format.
Response 6: The references have been revised.

Point 7: The conclusion is still poor.
Response 7: One conclusion was increased

Point 8: The paper still needs to increase the quality of the data to represent valuable information for readers, and still confirms the descriptions in the literature. The results and discussion are poor. The references need to be revised. I still suggest that the paper is not suitable for publication in Sustainability in this version.
Response 8: We thank you for your comments and hope that you will consider the modifications made.

Reviewer 4 Report

Dear Editorial office member

I checked the file in iThenticate and send it to editorial office by email here it. Similarity is 29%.

Other corrections are fine and just English should be improved more.

Regards

Author Response

Dear reviewer, thank you for your comment, the authors have corrected the similarity issue.

Thank you.

Reviewer 5 Report

Thank you for the opportunity to revisit an interesting topic „New characteristics in the fermentation process of cocoa (Theo-broma cacao L) "“Super árbol”" in La Joya de los Sachas, Ecuador“.

The Authors made a great job and did a lot of significant changes. However, further changes are necessary in order to recommend to accept the paper. I refer to the major concerns below.

Title: What "“Super árbol”" means?

Abstract: Well-written.

Keywords: “polifenoles“ must be changes to “polyphenols“.

Introduction: Well-written.

Materials and Methods

L 135-141: Las 100 habas de cacao tomadas al azar de las muestras y cortadas longitudinalmente por la guillotina se colocaron sobre una base blanca y se clasificaron en almendras bien fermentadas (cotiledones de color marrón
a marrón rojizo en toda su estructura y vetas de fermentación profundas), habas de cacao moderadamente fermentadas (cotiledones con un 50% de color marrón en su estructura), habas de cacao de color violeta (cotiledones de color violeta intenso) y habas de cacao de color pizarra (cotiledones de color gris negruzco con aspecto compacto) [20], [21]
.

Dear Authors, please note that writing a manuscript in two languages cannot take place. I suggest you must check the entire manuscript carefully. In addition, references must be quoted namely [20,21].

Statistical Analysis: This section must be overwritten. I think you must specify the statistical program with which you performed the statistical analysis in this paragraph. Additionally, please indicate which statistical methods have been used to verify the hypothesis of this study. The results must be reported in the Results section only.

Results

“Results and discussion” cannot be a single chapter. It seems necessary to distinguish between these subunits.

Table 3: It looks like the Authors have a requirement to correct values (for example, “9,75109E-07”, “0,000130369”) in the table. There are a lot of vague characters in the table that must be explained under this table.

Figure 3: Dear Authors, ANOVA does not show correlation or relationship. ANOVA stands for 'Analysis of variance' as it uses the ratio of between group variation to within group variation, when deciding if there is a statistically significant difference between the groups. For readers it will be impossible to understand the data located in Figure 3. Therefore, it is necessary to improve the resolution of this figure.

Discussion: Well-written. Nonetheless, I suggest that the Limitations paragraph can be presented at the end of the Discussion Unit.

Conclusions: Well-written

References: Must be improved according to MDPI rules.

Kind Regards

Author Response

Dear reviewer, we thank you for your comments and apologize for the oversights in the document. Each of the corrected errors allows us to improve. We, the authors, hope that this new version will be to your liking and that we can continue with the process and publish our manuscript.

Point 1: What "“Super árbol”" means?

Response 1: Super árbol means great tree, because farmers prefer this genetic material to Arriba and CCN51.

Point 2: “polifenoles“ must be changes to “polyphenols“.

Response 2: Super árbol means great tree, because farmers prefer this genetic material to Arriba and CCN51.

Point 3: L 135-141: Las 100 habas de cacao tomadas al azar de las muestras y cortadas longitudinalmente por la guillotina se colocaron sobre una base blanca y se clasificaron en almendras bien fermentadas (cotiledones de color marrón a marrón rojizo en toda su estructura y vetas de fermentación profundas), habas de cacao moderadamente fermentadas (cotiledones con un 50% de color marrón en su estructura), habas de cacao de color violeta (cotiledones de color violeta intenso) y habas de cacao de color pizarra (cotiledones de color gris negruzco con aspecto compacto) [20], [21].

Dear Authors, please note that writing a manuscript in two languages cannot take place. I suggest you must check the entire manuscript carefully. In addition, references must be quoted namely [20,21].

Response 3: We extend an apology for the oversight in language and references. In this revision, we have revised this aspect of the document.

Point 4: Statistical Analysis: This section must be overwritten. I think you must specify the statistical program with which you performed the statistical analysis in this paragraph. Additionally, please indicate which statistical methods have been used to verify the hypothesis of this study. The results must be reported in the Results section only.

Response 4: The statistical analysis section was reviewed and Spearman's correlation results were placed in Results.

Point 5: “Results and discussion” cannot be a single chapter. It seems necessary to distinguish between these subunits.

Response 5: This was corrected, item 3 only corresponds to Results and item 4 to Discussions.

Point 6: Table 3: It looks like the Authors have a requirement to correct values (for example, “9,75109E-07”, “0,000130369”) in the table. There are a lot of vague characters in the table that must be explained under this table.

Response 6: Table 3 was revised and corrected

Point 7: Figure 3: Dear Authors, ANOVA does not show correlation or relationship. ANOVA stands for 'Analysis of variance' as it uses the ratio of between group variation to within group variation, when deciding if there is a statistically significant difference between the groups. For readers it will be impossible to understand the data located in Figure 3. Therefore, it is necessary to improve the resolution of this figure.

Response 7: Figure 3 was revised and corrected

Point 8: Discussion: Well-written. Nonetheless, I suggest that the Limitations paragraph can be presented at the end of the Discussion Unit.

Response 8: Increased lines 365 - 368 as an addition to the last paragraph so that it can be considered as a "Limitations paragraph".

Point 9: References: Must be improved according to MDPI rules.

Response 9: References were reviewed and corrected

Round 3

Reviewer 1 Report

On table 3, p. 7, l. 217, please fix “Means scuares” to “Mean square”

Reviewer 5 Report

Dear Authors,

You answered my questions. This paper is interesting. I reccomend accepting it.

Kind Regards